Efficient-gastro: optimized EfficientNet model for the detection of gastrointestinal disorders using transfer learning and wireless capsule endoscopy images

Al-Otaibi Shaha 1
Rehman Amjad 2
Mujahid Muhammad 2
Alotaibi Sarah 3
Saba Tanzila tsaba@psu.edu.sa drstanzila@gmail.com 2
1 Department of Information Systems, College of Computer and Information Sciences, Princess Nourah bint Abdulrahman University , Riyadh , Saudi Arabia
2 Artificial Intelligence & Data Analytics Lab CCIS, Prince Sultan University , Riyadh , Saudi Arabia
3 Department of Computer Science, College of Computer and Information Sciences, King Saud University , Riyadh , Saudi Arabia
Sajid Ullah Syed
Electronic publication date: 2024 Mar 11
Publication date: 2024
Volume: 10
Electronic Location ID: e1902
Received 2023 Nov 21; Accepted 2024 Jan 31
Copyright: ©2024 Al-Otaibi et al.
Copyright year: 2024
Copyright holder: Al-Otaibi et al.
License: This is an open access article distributed under the terms of the Creative Commons Attribution License, which permits unrestricted use, distribution, reproduction and adaptation in any medium and for any purpose provided that it is properly attributed. For attribution, the original author(s), title, publication source (PeerJ Computer Science) and either DOI or URL of the article must be cited.
License URL: https://creativecommons.org/licenses/by/4.0/

Keywords: Deep learning, Gastrointestinal, Digestive endoscopy, Multiclass classification, Augmentation, Capsule endoscopy, Transfer learning

Funding: Princess Nourah bint Abdulrahman University, Riyadh, Saudi Arabia PNURSP2024R136 The Prince Sultan University, Riyadh Saudi Arabia This research is supported by Princess Nourah bint Abdulrahman University Researchers Supporting Project number (PNURSP2024R136), Princess Nourah bint Abdulrahman University, Riyadh, Saudi Arabia. The Prince Sultan University, Riyadh Saudi Arabia supported the APC of this publication. The funder had a role in the study design, data collection and analysis, decision to publish, and preparation of the manuscript.

==============================
Gastrointestinal diseases cause around two million deaths globally. Wireless capsule endoscopy is a recent advancement in medical imaging, but manual diagnosis is challenging due to the large number of images generated. This has led to research into computer-assisted methodologies for diagnosing these images. Endoscopy produces thousands of frames for each patient, making manual examination difficult, laborious, and error-prone. An automated approach is essential to speed up the diagnosis process, reduce costs, and potentially save lives. This study proposes transfer learning-based efficient deep learning methods for detecting gastrointestinal disorders from multiple modalities, aiming to detect gastrointestinal diseases with superior accuracy and reduce the efforts and costs of medical experts. The Kvasir eight-class dataset was used for the experiment, where endoscopic images were preprocessed and enriched with augmentation techniques. An EfficientNet model was optimized via transfer learning and fine tuning, and the model was compared to the most widely used pre-trained deep learning models. The model’s efficacy was tested on another independent endoscopic dataset to prove its robustness and reliability.

Introduction

Gastrointestinal diseases affect the digestive system and can lead to challenging medical conditions. Gastrointestinal conditions such as bleeding, ulcers, polyps, tumor cancer, colorectal cancer, etc. are extremely prevalent worldwide. Since radiologists and other medical professionals find it challenging to analyze large amounts of image data, there is a possibility for inaccurate diagnosis (Ling et al., 2021). Colorectal cancer is among the most common causes of mortality worldwide. Gastrointestinal cancer is the most prevalent form of cancer and the paramount cause of cancer-related death worldwide. According to the World Health Organization (WHO), gastrointestinal cancer is the fourth prevalent factor of death globally, affecting an estimated 1.8 million people each year (Sung et al., 2021). Polyps, which develop on the stomach and colon mucosa, are a key cause of gastrointestinal cancer. Polyps develop slowly and typically do not present any symptoms until they have reached a significant size (International Agency for Research on Cancer, 2018). We can prevent ourselves from developing polyps if they are diagnosed and treated promptly.

Gastrointestinal diseases are usually diagnosed via intestinal biopsy (IB). Medical professionals employ microscopic images to examine biopsy tissue samples for malignant or abnormal cells. According to Ishaque et al. (2021), intestinal biopsy method is invasive and requires a high level of expertise. Endoscopic imaging of the gastrointestinal tract is less invasive (Kainuma et al., 2015). Over the last two decades, endoscopy has revolutionized the diagnosis and treatment of many gastrointestinal diseases. Once again, endoscopy imaging is going through a significant revolutionary change. Diagnostic endoscopy’s emphasis is shifting from recognizing prominent disease to detecting subtle abnormalities. The endoscopic treatment helps the doctor find and treat gastrointestinal system abnormalities earlier. Chronic health issues may be addressed if detected and identified early (Ishaq et al., 2018). Because of this, the imaging process has the potential to significantly reduce medical problems, treatment costs, and death rates, particularly gastrointestinal cancer deaths. Gastrointestinal cancer is unavoidable if early warning signs are ignored or not addressed properly. Chemotherapy, radiation, and surgery are the most common treatments. Gastrointestinal cancer is more common in men (26%) than women (11%). Helicobacter pylori (H. pylori) Bacteria have the capability to invade the human body and establish habitation within the gastrointestinal system, causing ulcers and gastritis. Environmental and genetic factors can induce ulcers and gastritis (Korkmaz et al., 2017).

The world’s various healthcare systems could benefit from computer-assisted diagnosis now and in the future. A method that is computer-assisted and automated might be helpful in detecting polyps with a high degree of perfection in the initial phases of cancer. Methods that make use of artificial intelligence (AI) have shown a considerable degree of intimate in a range of medical fields for the purpose of aiding individuals in recognizing disorders that cannot be identified by the human eye by itself (Oka, Ishimura & Ishihara, 2021; Patel et al., 2021). Approaches that make use of artificial intelligence are able to discern between neoplastic and non-neoplastic tissues in a medical setting. The Kvasir collection of images of the gastrointestinal system includes two categories of endoscopic polyp removal, three therapeutically important outcomes, and three key anatomical landmarks. Endoscopy is now the most advanced method of gastrointestinal evaluation. When doing a colonoscopy, as opposed to a gastroscopy, which examines the stomach, ‘oesophagus’, and ‘upper portion of the small intestine’, the rectum and colon are both seen. Both identify and treat problems with the digestive tract. In each of the experiments, videos captured in real time and in high definition are utilized. The endoscopic equipment is less costly and requires specialized knowledge and training to operate properly (Golan et al., 2023).

In recent years, deep convolutional neural networks (DCNNs) have performed better than conventional machine learning models (El-Shafai et al., 2023; Mohapatra et al., 2023), especially in the recognition and classification of medical diseases. The efficacy of deep learning (DL) in this area can be attributed to its capacity to extract pertinent features using pre-trained models, which are subsequently improved to generate better results. In addition, transfer learning (TL) has proven to be a very important technique for training these pre-trained DL models because training methods that do not use TL are problematic due to resource limitations like memory and consumption time. In order to achieve even better levels of accuracy, researchers are currently focusing intensively on upgrading DL models by making them more complicated, combining numerous models, and fine-tuning their features. Deep learning has significantly changed the medical industry by making it possible to identify gastrointestinal tracts more accurately. This change has improved the models’ capacity to recognize mild symptoms by enabling them to detect complicated patterns and changes in endoscopic data in an effective manner. Some obstacles still exist, though, including the requirement for sizable and varied datasets, the possibility of an unequal distribution of classes, and the complexity of comprehending deep learning models in the context of medicine. Accurate disease diagnosis is essential in the medical to provide proper patient care and treatment. Deep learning techniques like VGG16, DenseNet, etc. are in high demand since traditional machine learning models have struggled to keep up with the ever-increasing complexity of medical data. Deep learning is able to automatically learn complex patterns and features from medical images, leading to improved accuracy in tasks involving the diagnosis and classification of different diseases.

Limitations and research gap

Advancements are being made in the classification of gastrointestinal disorders, despite the continued limitations associated with early-stage diagnosis, the scarcity of training datasets, and the need for high-quality imaging. Certain elements that contribute to such concerns are the complexity of the model design and the substantial processing demands. However, improvements to datasets, streamlining of model construction, and the development of methods accommodating varying degrees of image quality remain potential. Many studies combined different deep models that increased the computational time and were less accurate. The use of raw images for feature extraction has been observed to reduce classification accuracy. There is no proper augmentation techniques used to enhance the dataset and reduce the overfitting problems in the literature. The objective is to construct optimized deep learning models capable of early disease detection, potentially leading to improved patient outcomes and increased survival rates.

Research questions

This work aims to address three main research questions, and the answers are provided in the results and discussion section.

RQ1: What is the impact of choosing and integrating data augmentation approaches on the performance of the optimized EfficientNet model for gastrointestinal disease detection?

RQ2: How effective is the model in terms of complexity and processing speed?

RQ3:: Is the model robust to different datasets.

Research motivation

The gastrointestinal system is essential for the identification and treatment of a number of diseases, such as stomach and esophageal malignancies. The gastrointestinal images in the Kvasir collection represent an important area of study that has the potential to greatly enhance medical practices and healthcare facilities worldwide. For effective treatment and ongoing surveillance, the diagnosis of gastrointestinal cancers—in particular, colorectal cancer—is essential. However, medical professionals’ proficiency in detecting a condition may have an impact on the precision of the diagnosis. Enhancing the evaluation and detection of gastrointestinal disorders using automated diagnostics might result in higher productivity and better use of available resources. The availability of extensive medical databases and technological advancements in healthcare procedures has greatly improved the diagnosis and treatment of a wide range of disorders. The proficiency of endoscopists and factors related to the gastrointestinal tract compromise detection accuracy. It is thought that this investigation’s findings will be very important in lowering the number of gastrointestinal cancer cases and fatalities.

Research contributions

The research contributed the following points:

• We proposed an EfficientNet method with fine-tuned Conv2D layers and Dropout layers to accurately and efficiently detect and classify gastrointestinal disorders, utilizing two datasets that are available to the public. One dataset has eight diseases, and the second dataset has four gastrointestinal-related diseases. The main aim of this method is to build a lightweight, reliable, and less parameter-based approach that makes detection easier.

• The collected dataset is then augmented to enhance the number of training images for deep learning models and preprocessed to enhance its quality.

• The pretrained deep learning models are used for the Detection of gastrointestinal tract disorders, and previous handcrafted methods are compared with the proposed approach. Accuracy, precision, recall, F1 measure score, AUC, loss, and true and false predictions are all used to assess the significance. Also, a detailed analysis is performed.

• To address the data overfitting problem and demonstrate the model’s effectiveness, this study utilized cross-validation experiments. The accuracy of the proposed method outperforms that of other state-of-the-art methods.

The subsequent sections of the article are structured as follows. ‘Related work’ offers a summary of the research that has been conducted on the detection of gastrointestinal disorders through the use of wireless endoscopic data sets. In ‘Proposed methodology’, the methodology that is offered is described in detail. This technique involves the preprocessing of the dataset, dataset augmentation, the implementation of the proposed method, and the evaluation of the effectiveness of the proposed method using performance metrics. Results and discussion are presented in ‘Results and discussion’. The conclusion and recommendations for further research are presented in ‘Conclusion and future work’.

Related Work

Many scholars have developed multiple methods, all of which utilize endoscopy images, to diagnose the cancer. Deep learning techniques have already been utilized in medicine for the detection of multiple diseases, such as employing neural networks to classify stomach cancer, and deep learning (Majid et al., 2020) to classify stomach abnormalities. Escobar et al. (2021) developed a method that outperformed existing techniques for diagnosing diseases and anomalies affecting the gastrointestinal tract using endoscopic images. The main focus of the proposed approach was transfer learning through the utilization of the VGG16 network, which had been pre-trained on the ImageNet task. Wang et al. (2022) developed classification approach integrating convolutional neural network (CNN) and capsule networks to automatically detect gastrointestinal tracts. The authors reported that the proposed network achieved classification accuracy of 94.834% in the Kvasir dataset and 85.995% in the Hyperkvasir dataset. Hosain et al. (2022) employed transfer learning to identify esophagitis, polyps, and ulcerative colitis alongside healthy colon images using DenseNet201 and a vision transformer. DenseNet201 was outperformed by Vision Transformer, which had an accuracy of 95.63% as opposed to DenseNet201′s 71.88%. The authors claim that they used a limited dataset.

Noor et al. (2023) proposed a deep learning-based architecture for correctly classifying gastrointestinal tract disorders. The major goal was to underline the importance of this process for endoscopy images while enhancing the utilization of fully adjustable brightness. Different performance measurements were used to assess the performance of contrast enhanced-images. Additionally, data augmentation was carried out to expand the dataset and improve the model’s generalizability. Then, used transfer learning technique, a pre-trained model was adjusted for endoscopic purposes, followed by the utilization of several machine learning methods to implement the retrieved features of the improved model. By employing the ‘softmax’ classifier, the proposed model achieved overall accuracy rate of 96.40%, a recall rate of around 93%, a precision rate of nearly 97%, and overall F-measure of 95.24%.

A unique CNN and long short term memory (LSTM) based diagnosis methodology was recommended for the classification of various gastrointestinal conditions employing endoscopic recordings. Furthermore, their recommended classification methodology was utilized to develop an outcome based endoscopy video retrieval method. In comparison to features learned only from spatial data, the proposed temporal feature-based technique encodes better discriminate interpretations of numerous endoscopic images. As a result, the effectiveness of classification and retrieval was enhanced by spatial and temporal data. The model’s results were thoroughly analyzed using the Kvasir database. Additionally, a comparison of the various other recent approaches was conducted using same database and experimental methodology. The proposed method yielded an overall accuracy of 92.57% (Owais et al., 2019). Fati, Senan & Azar (2022) demonstrated a classification driven retrieval method based on deep learning that can be used to identify various categories of gastrointestinal diseases. The entire structure consists of a deep learning-based classification network and an identification technique. The classification network anticipates the type of disease based on the present state of health and its regaining component then displays cases from the preceding database that match the predicted disease.

A significant amount of research had been conducted in the era of medical imaging to aid medical professionals in accurately identifying and classifying medical diseases such as lung cancer, breast tumors, and brain tumors (Lahoura et al., 2021; Fahad et al., 2018), as well as stomach-related infections, utilizing endoscopy images (Naz et al., 2021). The stomach performs a vital function within the human body. Harmful conditions of the stomach include hemorrhaging, polyps, ulcers, and gastroenteritis. Khan et al. (2019) developed a novel method with the goal of identifying and classifying abnormalities in gastrointestinal disease. Kvasir and private databases were used to evaluate the entire method. The proposed strategy demonstrates greater success than the currently employed methods.

The authors utilized various preprocessing techniques and fine-tuned the MobileNetV2 model for gastrointestinal disease detection, while also employing Grad CAM and ResNet152 for a comprehensive deep learning technique for endoscopic image classification. They made use of the 8,000 wireless capsule images that were available for viewing in the freely available Kvasir database (Khan et al., 2022). A high-performing outcome for the classification of medical images was achieved by using an efficient augmentation method in conjunction with the classification results, which had an accuracy of 98.28% during training and 93.46% during validation (Mukhtorov et al., 2023). In a separate piece of research, the researchers explain the methodologies and processes for applying deep learning algorithms to examine a wide variety of gastrointestinal disorders and recognize these images. On the Kvasir dataset, they retrained five cutting-edge neural network architectures, including ResNet, MobileNet, Inception-v3, and Xception, to classify eight distinct classifications. These categories include anatomical landmarks, diseases, and surgical procedures. When compared to methodologies that are now considered to be state-of-the-art, the findings that their models had produced exhibited an astounding level of performance, with results that were accurate and showed potential. The accuracy rates that were acquired through the use of VGG, ResNet, MobileNet, and Inception-v3 were, respectively, 98.3%, 92.3%, 97.6%, 90%, and 98.2%. Retraining the VGG16 and Xception neural networks appears to have provided the most accurate results, with accuracy exceeding 98% as a consequence of their outstanding training efficiency on the ImageNet dataset and internal structure that supports classification difficulties. This accuracy was achieved as a result of the combination of these two factors (Dheir & Abu-Naser, 2022).

The existing work on diagnosing gastrointestinal tract diseases is still developing, despite its accuracy. However, there is still potential for improvement due to insufficient endoscopic images, incorrect evaluation criteria, and a focus on a single disease. Accuracy is a crucial performance metric, but its evaluation in multi-class classification has received less attention, especially when dealing with imbalanced data sets.

Previous studies have highlighted the limitations of deep learning models, including the time and computing resources required for training large datasets. However, our research successfully overcomes these limitations by using an optimized version of the EfficientNet model for classification. The dataset was preprocessed and improved using augmentation strategies like magnification, random contrast, and brightness modifications. The EfficientNet model was introduced using transfer learning and fine-tuning approaches. The model was compared to pre-trained deep learning models and evaluated on a different endoscopic dataset to ensure its longevity and reliability. This optimized learning approach addresses the limitations of deep learning models on enormous datasets. The comparison of limitations and contributions to related work is shown in Table 1.

Table 1 Comparison of limitations and contributions on related work.

Methods	Datasets	Limitations	Contributions	
CNN (Wang et al., 2022)	Kvasir	CNNs are unable to learn global features, which are crucial for classifying endoscopic images of the gastrointestinal tract. Also, enhancement techniques were not employed for accuracy.	Their approach involves utilizing a CNN feature extraction unit for saving specific details about lesions into midlevel CNN features. This allows capsule classification networks to effectively learn associations between image elements that are invariant to deformations.	
VGG-16 (Escobar et al., 2021)	Kvasir-V2	As a result of using pre-trained approaches, the authors were unable to attain satisfactory detection performance. The amount of time required for computation is also large.	The contributions of this study are to address the classification issue utilizing the Kvasir-V2 dataset by employing pre-processing techniques, transfer learning methods, and hyperparameter tuning. In order to tackle the issue of having a limited number of annotated medical shots, data augmentation techniques are employed to create extra images through geometric modifications.	
DenseNet201 (Hosain et al., 2022)	WCE	The authors did not utilize the computational time, area under the curve score, and other measures, resulting in low accuracy.	Researchers used ViT and DenseNet201 models to detect three gastrointestinal illnesses and a healthy colon from wireless capsule endoscopy images.	
MobileNet-V2 (Noor et al., 2023)	Kvasir	This study did not consider the consequences of training numerous models and integrating them, and the optimization of features, resulting in low precision.	This study presents an evolutionary algorithm-based method for maximizing contrast enhancement controlled by brightness. It uses a pre-trained deep CNN model for extracting and classifying significant characteristics. A quantitative study was conducted to evaluate the effectiveness of the proposed technique in improving the classification of gastrointestinal illnesses using performance measures.	
CNN+LSTM (Owais et al., 2019)	Kvasir	The authors employed a hybrid model with multiple layers, resulting in increased computational cost and delayed decision-making.	The authors suggest a unique architecture that outperforms earlier techniques based on manually created features and 2D-CNNs for learning the spatial and temporal aspects of different gastrointestinal diseases. This design uses ResNet and LSTM.	
MobileNet-V2 (Khan et al., 2022)	Kvasir	The classification process’s error rate increases due to the removal of irrelevant characteristics, resulting in poor segmentation of infected regions.	The study indicates a method that combines different techniques to improve the visibility of infected regions in medical image segmentation. This method involves using a deep learning technique called a saliency-based approach, fine-tuning a MobileNet-V2 model, and utilizing a hybrid algorithm for selecting the most relevant features.	
VGG-16 (Dheir & Abu-Naser, 2022)	Kvasir	The authors primarily focused on the dataset classes, without providing a thorough explanation of the proposed framework or its architectural specifics.	They examine several approaches and methodologies for utilizing deep learning algorithms in the study of computer-aided detection of multiple diseases in the gastrointestinal tract, specifically focusing on image identification.	

Proposed Methodology

The authors proposed the EfficientNet-Gastro method with fine-tuned Conv2D layers and Dropout layers to accurately and efficiently detectand classify gastrointestinal disorders using two publicly available datasets. Figure 1 demonstrates the proposed methodology. The authors obtained a gastrointestinal disease image dataset that contains images with different sizes, like 900* 1040, 1024* 768, etc. During processing, each model required a fixed-size dataset. So, we performed preprocessing on the Kvasir dataset. After preprocessing, the authors, to increase training data,augmented the data using the augment library. Then apply deep learning and the proposed model with fine-tuning and transfer learning methodologies to minimize the computation time. A step-by-step description of the proposed methodology is presented in the subsection below.

Figure 1 Gastrointestinal flow diagram.

The gastrointestinal disease image dataset, despite its size variations, was preprocessed and augmented for deep learning, and a transfer-based model was proposed for classification.

Preprocessing and data augmentation

The main theme of pre-processing is to enhance the quality of an image in order to facilitate more effective evaluation. Preprocessing techniques are employed to minimize undesirable distortions and enhance particular features that are crucial for the desired result. Preprocessing is essential to resize the original images in the dataset into similar or fixed sizes for the convenience of deep models because the original images in the dataset are in various shapes and dimensions (Mujahid et al., 2022). We resized the images into equal sizes using the Python module of computer vision. Additionally, image preprocessing can reduce model training and inference times. when the images being used are especially large, reducing their size will significantly decrease model time for training without substantially affecting the model’s performance.

Figure 2 shows the samples of gastrointestinal tacts. Augmentation is the process of creating new samples from existing samples in a dataset to enhance the dataset, and preprocessing is the process of adjusting existing images to satisfy certain requirements. The Image Augmentor Python package is deployed to enhance the data class images during image augmentation process. Algorithm 1 details the process of data enhancement. The rotation is the first stage in a series of phases that make up the data augmentation process. The zooming, flip_left_right, and flip_top_bottom operations come next. Last but not least, we employ a sample rate and the random_distortion function. In order to prevent model overfitting for data from the majority class, new images are produced to balance the dataset. Splitting the augmented dataset into 90:10 ratio is illustrated in Table 2.

Figure 2 Sample of various gastrointestinal diseases.

Table 2 Splitting the augmented dataset into 90:10 ratio.

Disease label	Training_samples	Testing_samples	Total_samples	
dyed lifted polyps	2,091	232	2,323	
dyed resection margins	2,097	233	2,330	
esophagitis	1,966	218	2,184	
normal cecum	2,010	223	2,233	
normal pylorus	2,078	231	2,309	
normal z line	1,967	219	2,186	
polyps	1,967	219	2,186	
ulcerative colitis	2,024	225	2,249	
Total_samples	16,200	1,800	18,000	

_______________________ Algorithm 1 Augmentation._________________________________________________________________ Input:_ Training Images Output:_ Augmented Dataset import Augmentor Start:   1:  Aug_pipeline ← Augmentor.Pipeline(”images_path”)   2:  Aug_pipeline ← (Zoom_probability = 30%,percentage_area = 40%)   3:  Aug_pipeline   ←  (Rotate_robability   =    30%,Max_left_rotation   =      5,Max_right_rotation = 5)   4:  Aug_pipeline ← Flip_Left_Right = 30%)   5:  Aug_pipeline ← Flip_Top_Bottom = 30%)   6:  Aug_pipeline  ← Random_distortionprobability  =   40%,Gridwidth  =      2,Gridheight = 2,magnitude = 7) End__________________________________________________________________________________________________

Proposed method

EfficientNet-B1 is a part of the family of EfficientNet and was created by the researchers at Google in 2019. The lowest computational time, efficiency, model size, and high perfromance and effective architecture are the power of this EfficientNet family. The proposed methodology incorporates the utilization of a CNN architecture and a scaling method that employs a compound coefficient approach. This approach ensures the consistent scaling of all parameters related to depth, width, and resolution. The EfficientNet scaling strategy involves a systematic increase in the width, depth, and resolution of the network using a predefined set of scaling coefficients. This differs from the conventional approach, which often scales these elements independently, potentially resulting in a less robust model. The compound scaling strategy is justified by the requirement to accommodate larger input images by incorporating extra layers. This allows for an improved receptive area within the network, enabling the identification of more significant elements over the entire image. Additionally, the inclusion of more channels further enhances the network’s ability to detect key features. EfficientNet provides high accuracy and is specially adopted in such a case where there’s limited memory and resources. It works efficiently and achieves outclass results in computer vision and classification-related tasks. Figure 3 displays the proposed method’s architecture.

Figure 3 The proposed architecture for gastrointestinal disease detection.

The authors used the EfficientNet architecture with the transfer learning process because building an architecture from the initial stages is more challenging and time-consuming. Transfer learning is a technique in which learned knowledge is again used for another task by fine-tuning the parameters and modifying the additional layers. To minimize the total parameters and overfitting, the authors implemented a global pooling average layer to the EfficientNet model. A number of dense layers with dropout layers and relu activation functions was also used. To avoid overfitting, a dropout rate of 35% was chosen at random. A softmax activation function was used to one output dense layer with four output units for multiclass classification in dataset 1 and eight output units in dataset 2 build the proposed automated model. Also, two Gaussian noise layers with a 25% rate each were added.

Throughout the experiments, open-source software and libraries are utilized. To speed up the training, the authors used Google Colab Notebook with GPU run time. In simple terms, the software utilized has no license constraints since it is both open source and free to use.

_______________________________________________________________________________________________________ Algorithm 2 Algorithm for the Proposed method.____________________________________ base_model = EfficientNet weights = ’Imagenet’ input_shape = (224 * 224 *3) Mo = model Start:   1:  Mo = Sequential()   2:  Mo ← base_model  3:  Mo ← Conv2D(128,kernelsize = (3,3),activation =′ relu′)   4:  Mo ← Conv2D(512,kernelsize = (3,3),activation =′ relu′)   5:  Mo ← GaussianNoise()   6:  Mo ← GlobalAveragePooling2D()   7:  Mo ← Dense(512,activation =′ relu′)   8:  Mo ← BatchNormalization()   9:  Mo ← Dropout() 10:  Mo ← Dense(512,activation =′ relu′) 11:  Mo ← BatchNormalization() 12:  Mo ← GaussianNoise() 13:  Mo ← Dense(8,activation =′ softmax′) End__________________________________________________________________________________________________

Dropout regularization is a technique that is similar to training multiple neural network architectures simultaneously. A subset of layer results are arbitrarily “dropped out” or neglected during training. Any change made during training utilizes a distinct “view” of the layers in its final state. Dropout layers are very crucial to in the training process for deep learning models as they prevents from overfitting the data. The first set of samples for training has an abnormally high level of effect on the learning process and thus prevents the learning of features that appear only in later batches. Batch normalization is a method of supervised learning that normalizes the intermediate outputs of a neural network by converting them into a standard structure. This process is also known as normalizing. This “resets” the distribution order of the output from the layer previous it so that the layer above it can handle it more effectively. After removing the mean batch and scaling by the standard deviation, the output of the preceding activation layer is transformed by batch normalization. This makes a deep learning network more stable. Table 3 represent model architecture and their parameter values for gastrointestinal disease detection.

Table 3 Model architecture and their parameter values for gastrointestinal disease detection.

Layers	Output	Parameter values	
Functional (EffiecientNet-B1)	(7,7,1280)	6,575,,239	
Conv2D	(95,5,128)	1,474,688	
Conv2D	(3,3,512)	590,336	
GaussianNoise	(3,3,512)	0	
GlobalAveragePooling2D	(512)	0	
Dense	(512)	262,656	
BatchNormalization	(512)	2048	
Dropout	(512)	0	
Dense	(512)	262,656	
BatchNormalization	(512)	2048	
GaussianNoise	(512)	0	
Dense	(8)	4104	

Results and Discussion

In this section, the experimental results acquired with multiple DL models and the proposed model using the Kvasir eight class dataset and the four-class wireless capsule endoscopy dataset are discussed.

Experimental setup

The efficacy of the proposed approach was assessed by employing Google Colab and two datasets sourced from the Kaggle repository. The equipment that was utilized comprised an Intel Core i7-7500U central processing unit, 32 GB of random-access memory, 24 GB of video memory capacity, and an NVIDIA GeForce RTX 3080 graphics card. TensorFlow and PyTorch were employed to enhance computing efficiency. The selection of Python 3.10 was based on its comprehensive library and adaptable nature. The study examined augmentation and preprocessing techniques and compared the model’s accuracy, training duration, and detection speed with previous studies. The datasets were partitioned into two distinct groups for analysis: 90% for the purpose of training and 10% for the purpose of testing. The study seeks to make a substantial contribution to the field of medical image processing, specifically by focusing on the identification of gastrointestinal disorders. Deep learning technology integration has the potential to enhance diagnosis accuracy, resulting in substantial treatment advantages. The timely and accurate detection of gastrointestinal problems holds great clinical importance. Table 4 presents the training configuration of proposed Model architecture.

Table 4 Training configuration of proposed model architecture.

Optimizer	Adam	
Batch size	64	
Learning rate	0.0001	
Epochs	25	
No of features	8	
Execution time	1100 s	
Trainable parameters	2, 596, 488	
Loss	Categorical cross entropy	

Performance metrics

Classification problems are among the subjects that have received the greatest research attention everywhere on the globe. Use cases may be found in almost every medical setting. Identifying different writers, voices, and the content of written words. We need a metric that compares discrete classes in some way in order to make use of the fact that the output from classification models is discrete. Classification metrics evaluate the performance of a model and provide information on how accurate the classification is; however, they accomplish this in a variety of different ways (Padilla, Netto & Da Silva, 2020).The ratio of correct predictions relative to the total number of predictions, multiplied by 100, is called classification accuracy. This may be the statistic that is easiest to understand and use. The authors employed the abbreviation “_g” in conjunction with the metrics formulas, wherein “_g” represents the gastrointestinal.

Accuracy in classification models is a metric that quantifies the proportion of accurately predicted instances. It is determined by dividing the number of correct predictions by the total number of predictions made. (1) Accuracy_g=TP_g+TN_gTP_g+TN_g+FP_g+FN_g∗100

Precision is the capacity of a model to reliably predict positive results. The ratio of accurate positive predictions to erroneous positives determines it. High accuracy signifies the effective mitigation of false positives, particularly in contexts such as medical diagnosis. (2) Precision_g=TP_gTP_g+FP_g∗100

Recall is an essential classification parameter that measures a model’s capacity to accurately identify as well as incorporate true positives. Optimizing recall is essential in medical diagnostics since the failure to detect positive instances might result in significant repercussions. (3) Recall_g=TP_gTP_g+FP_g∗100

An important classification metric that balances recall and accuracy to provide a fair assessment of a model’s utility is the F1-score. It strikes a compromise between memory and accuracy, particularly when obtaining real positive instances and lowering false positives. It is frequently applied to assess the overall performance of the model. (4) F1−score_g=2∗pre_g+Rec_gPre_g∗Rec_g∗100

In evaluating the efficacy of classification models, the AUC, which represents the area under the ROC curve, is a crucial metric. An AUC value exceeding 1 indicates perfect categorization, while a value below that indicates superior discrimination. It provides a comprehensive viewpoint regarding the sensitivity and specificity of a model. (5) AUC_g=0.5∗TPR_g+TNR_g∗100.

Datasets

Automated identification of multiple diseases through the use of technology such as computers is a significant yet unexplored area of research. These innovations might enhance healthcare services and improve global health care networks. However, databases with medical images are limited, which makes reproducibility and method comparison nearly impossible. In this article, we provide the Kvasir dataset, which contains images of the gastrointestinal diseases. The dataset has been arranged and annotated by medical professionals (Expert Endoscopists). Norway’s Vestre Viken Health Trust (VV), which is endowed with endoscopic technology, collects the data. TThe VV consists of 4 hospitals and offers medical services to a population of 470.000 individuals. Baerum Hospital, one of the listed medical facilities, has a significant department specializing in gastrointestinal medicine, which served as the primary source for the acquisition of training data.

The KVASIR dataset (Pogorelov et al., 2017) provides a comprehensive compilation of endoscopic operations across the gastrointestinal (GI) tract, encompassing a diverse range of polyps. This dataset is the initial one to faithfully replicate these techniques, with each category including 1,000 images. Eight different types of abnormalities are separated in the dataset: normal-cecum, normal-pylorus, normal-z-line, esophagitis, polyps, ulcerative colitis, dyed-lifted polyps and dyed-resection-margins. Table 5 shows the specific description about the dataset.

Table 5 Specific description about the dataset.

Disease label	Description	
normal-cecum	The efficacy of colonoscopy is contingent upon the presence of the normal cecum, a segment situated at the commencement of the large intestine that can be distinguished by the appendix access. To validate intubation, a conventional configuration of an electromagnetic scope is employed. The size of the appendix opening and the configuration of the scope used both affect how accurate the results are from a colonoscopy.	
normal-pylorus	The pylorus, which encircles the duodenum, is important for controlling the passage of food from the stomach to the small intestine. The circumferential muscles help it accomplish this. Using endoscopic duodenal instrumentation, a sophisticated technique used in gastroscopy, requires an accurate location of the pylorus. The goal of a full gastroscopy is to look for any anomalies, such erosions, ulcers, or constriction of the stomach on both sides.	
normal-z-line	The Z-line, which appears as a clear division between the white and red mucosa, marks the border between the stomach and esophagus. To ascertain if a disease is present and to characterize the pathology of the esophagus, it is imperative to recognize and assess this line.	
esophagitis	Esophagitis is an inflammation of the esophagus that impacts the Z-line and is caused by a mucosal breach. Frequent causes include gastric reflux disease, hernias, and regurgitation. The extent of inflammation can be inferred from the dimensions and duration of the mucosal tears. Computer detection systems have the capability to generate automated reports and aid in the assessment of severity.	
polyps	Intestinal polyps can be flat, elevated, or stalk-like and distinguishable from normal tissue. Some entities may initially be harmless, but some can become cancerous. Colorectal cancer prevention requires early detection and removal. Automatic identification might improve polyp diagnosis, evaluation, and reporting.	
ulcerative colitis	The diagnosis of ulcerative colitis, an ongoing inflammatory condition that impacts the large intestine, is established through colonoscopy examination. It impacts quality of life and necessitates precise evaluation.	
dyed-lifted polyps	The risk of electrocautery causing harm to the lower layers of the gastrointestinal (GI) wall is minimized when polyps are elevated. It is essential to identify the locations where polyps can be excised.	
dyed-resection-margins	The evaluation of resection margins is crucial for determining the full excision of the polyp. Remaining polyp tissue can result in ongoing proliferation and, in the most severe scenario, the formation of malignancies.	

The proposed framework is also evaluated using the second dataset (Silva et al., 2014). The dataset has four distinct classifications, namely ulcerative colitis, polyps, normal, and esophagitis. There are 1,000 instances in each class. The examination technique at WCE has been greatly enhanced, leading to improved findings while ensuring patient comfort. The device provides a visual representation of the gastrointestinal system, resembling conventional techniques, and its compact and capsule-shaped design aids medical professionals in identifying issues with more efficiency.

Performance of pretrained deep learning models

The experimental results of fin-tuned pretrained deep learning models are listed in Table 6. In all medical related disease detection, artificial intelligence based deep learning models are attained outclass accuracy as compared to traditional or handcrafted machine features. DenseNet helps solve the problem of vanishing gradients, improves feature transfer, promotes utilizing features, and significantly reduces the total number of para-meters. DenseNet121 used to the Kvasir dataset in order to detect gastrointestinal tracts. The Densenet121 detects the gastrointestinal tracts with an overall and micro-average accuracy of 88% and 89%, respectively. This model achieved a 91% precision and 86% F-measure score for esophagitis. The ulcerative-colitis class achieved 98% highest precision and 92% F-measure. The CNN model has an overall accuracy of 95.22% and a polyps accuracy of 96%. The VGG16 model attained an accuracy of 97.6% and an area under the curve of 99.54%. In comparison to CNN and DenseNet models, VGG16 achieved 98% accuracy on polyps. ResNet50 also achieved great results, while Xception achieved lowest results. Figure 4 presents the performance metrics of pretrained models.

Table 6 Performance of pretrained deep learning models.

Methods	Disease	Precision	Recall	F1 measure	AUC	Loss	Acc	
DenseNet121	dyed lifted polyps	0.75	0.94	0.83	99.17	0.31	88.11	
dyed resection margins	0.93	0.70	0.80	
esophagitis	0.91	0.81	0.86	
normal cecum	0.93	0.93	0.93	
normal pylorus	0.95	1.00	0.97	
normal z line	0.83	0.91	0.87	
polyps	0.84	0.89	0.86	
ulcerative colitis	0.98	0.88	0.92	
CNN	dyed lifted polyps	0.91	0.95	0.93	98.27	0.10	95.22	
dyed resection margins	0.95	0.91	0.93	
esophagitis	0.95	0.98	0.97	
normal cecum	0.94	1.00	0.97	
normal pylorus	0.98	1.00	0.99	
normal z line	0.98	0.94	0.99	
polyps	0.96	0.89	0.92	
ulcerative colitis	0.95	0.95	0.95	
VGG16	dyed lifted polyps	0.99	0.97	0.98	99.54	0.12	97.61	
dyed resection margins	0.98	0.99	0.99	
esophagitis	0.99	0.90	0.94	
normal cecum	1.00	0.99	1.00	
normal pylorus	0.97	1.00	0.98	
normal z line	0.91	0.98	0.94	
polyps	0.98	0.99	0.99	
ulcerative colitis	0.99	0.98	0.99	
ResNet50	dyed lifted polyps	0.99	0.97	0.98	99.62	0.12	96.61	
dyed resection margins	0.97	1.00	0.98	
esophagitis	0.87	0.98	0.92	
normal cecum	0.98	0.98	0.98	
normal pylorus	0.99	1.00	0.99	
normal z line	0.97	0.85	0.91	
polyps	0.99	0.96	0.97	
ulcerative colitis	0.98	0.99	0.98	
Xception	dyed lifted polyps	0.80	0.66	0.72	97.19	0.62	77.94	
dyed resection margins	0.71	0.89	0.79	
esophagitis	0.83	0.79	0.81	
normal cecum	0.97	0.62	0.75	
normal pylorus	0.75	0.98	0.85	
normal z line	0.83	0.63	0.72	
polyps	0.72	0.75	0.74	
ulcerative colitis	0.75	0.91	0.82	

Performance of proposed model using the Kvasir and WCE datasets

The performance of the proposed model using Kvasir and the WCE dataset is presented in Table 7. The proposed model achieved 98.94% accuracy using the Kvasir 8 eight-class dataset and 96.63% accuracy on the WCE dataset. The proposed model obtained the 99.87% maximum AUC score using the Kvasir dataset. All diseases achieved micro-average scores of 99%. The results of the proposed technique is exceptional across all eight classifications. Using kvasir data, only patients with normal-cecum disease obtained a recall rate of 95%, whereas all others achieved a minimum of 97% and a maximum of 100% accuracy. The authors also conducted additional experiments utilizing an additional WCE dataset, and the proposed model obtained an AUC score of 99.58%.

Figure 4 Overall performance metrics of several pretrained models.

X-axis presents the deep models and Y-axis presents the performance analysis.

Table 7 Performance of proposed model using the Kavasir and WCE datasets.

	Disease	Precision	Recall	F1 measure	AUC	Loss	Acc	
Dataset1	dyed lifted polyps	1.00	1.00	1.00	99.87	0.03	98.94	
dyed resection margins	1.00	1.00	1.00	
esophagitis	1.00	0.97	0.98	
normal cecum	1.00	0.95	0.97	
normal pylorus	1.00	1.00	1.00	
normal z line	0.98	1.00	0.99	
polyps	0.98	1.00	0.99	
ulcerative colitis	0.97	1.00	0.98	
Dataset 2	normal	0.99	1.00	1.00	99.58	0.11	96.63	
ulcerative colitis	0.96	0.91	0.94	
polyps	0.94	0.95	0.95	
esophagitis	0.97	1.00	0.99	

The efficacy of a classification model can be measured using a confusion matrix. If there are unequal numbers of tests in each class is imbalanced or if the data has more than two classes, classification accuracy by itself can be inaccurate and misleading. By creating a confusion matrix and examining the results, authors might be able to better understand the positive and negative aspects of your classification model. The matrix demonstrates the values of true positives (TP), true negatives (TN), false positives (FP), and false negatives (FN) that were created by the model based on the test or validation data. The confusion matrix is utilized in order to determine the number of true add false predictions for each class. These figures are then organized according to the class that was predicted. In addition, the confusion matrix may be used to generate several metrics, including as recall, precision, and F1 measure, that are employed to analysed the performance of classification models.

Figure 5 illustrates the results of the proposed model using the confusion matrix. In the confusion matrix, horizontal illustrates the actual classes and vertical shows the predicted classes. In the figure, 0 indicates dyed_lifted polyps, 1 dyed_resection margins, 2 esophagitis, 3 normal_cecum, 4 normal_pylorus, 5 normal z_line, 6 polyps, and 7 ulcerative_colitis disease. Class zero makes 99.1% correct predictions, class one makes 100%, class two makes 97.7%, classes three and four make 99.6%, class five makes 99.5%, class 6 makes 98.2%, and class seven makes 100% correct predictions. The most wrong predictions made by the class two mean esophagitis disease, and the most correct predictions were made by the classes one and seven. The proposed model overall makes 960 correct predictions.

Figure 5 Results of proposed model using confusion matrix.

X-axis shows the true class and Y-axis shows the predicted class. Class 1 means dyed resection margins disease, and Class 7 means ulcerative colitis disease has the most accurate predictions.

Plots that are referred to as learning curves are utilized in order to demonstrate how well a model performs on training set. The training loss metric is used as an indicator of how well a deep learning model fits the training data. To put it another way, it determines how inaccurate the model is based on the data from the training set. Deep model was constructed using only a portion of the dataset, which is referred to as the training set. The amount of training loss can be computed by adding up all of the errors made by each training set sample. Additionally, it is essential to keep in mind that the training loss will be measured at the conclusion of each batch. The training loss curve may often be seen by plotting it, which is one typical way. For the purpose of determining how well a deep learning model operates on a validation set, a metric known as validation loss is utilized. A portion of the dataset known as the validation set has been separated off so that the accuracy of the model can be assesses. The validation loss, which is related to the training loss, is calculated by adding up the number of errors made across all of the samples that are included in the validation set.

These types of curves are essential for evaluating the performance of models on training or validation sets. Figure 6 depicts a variety of learning curve graphs utilizing pretrained models and the proposed model. We have plotted the training and validation curves for five pretrained deep learning models. We observed that the proposed model had the maximum accuracy and lowest loss compared to all other models shown in Fig. 6.

Figure 6 Accuracy and loss learning curves of several pre-trained deep learning and the proposed method.

The training process involves a model’s performance fluctuating at each epoch, indicating its progress in uncovering new information. Initial accuracy increases and loss decreases, but fluctuations occur when encountering unfamiliar data. Monitoring precision and mistake ratios enhances deep models’ resilience and effectiveness on unfamiliar data.

Deep learning models are intrinsically more dependent on processing resources than other techniques. This is due to the fact that deep learning models contain a massive number of parameters and demand a larger amount of training data. It is necessary to calculate how much time will be needed to run the model in the training procedure. Calculating the computational time of deep learning models versus the proposed model is the main emphasis of this study. The computational time of any model is a challenge that must be solved by all researchers working in this field in the modern era. The researchers are able to better determine the relevance of each hyper-parameter they choose for the creation of a model with the assistance of time complexity. Table 8 shows the computational time and loss of the proposed model and the pretrained deep learning models to differentiate how efficient the proposed model is computationally. The computational time of the xception model is 72 s per epoch, and it achieved a 0.62 validation loss, which is the highest among other models. The proposed model achieved a 0.03 validation loss and takes 44 s per epoch.

Table 8 Computational time of Proposed method.

Models	Consumption time	Training loss	Validation loss	
DenseNet121	50 s per epoch	0.2024	0.3059	
CNN	38 s per epoch	0.0242	0.102	
VGG16	69 s per epoch	0.0347	0.1195	
ResNet50	56 s per epoch	0.2432	0.1232	
Xception	72 s per epoch	0.2982	0.6236	
Proposed Model	44 s per epoch	0.0147	0.0381	

Figure 7 illustrates the loss caused by the proposed model in comparison to the pretrained deep learning models, hence providing ways to assess the relative efficiency for the current method. The Xception model demonstrates a validation loss of 0.62, surpassing that of other models. The proposed model attained a validation loss of 0.03.

Figure 7 Overall training and validation loss of the proposed method vs. other pretrained models.

X-axis indicate the models and Y-axis indicate the loss values.

The authors also performed experiments using k-fold cross-validation as presented in Table 9. The authors performed twofold, fivefold, and tenfold validation on the proposed model to check the model’s efficiency. The authors achieved 97.04% accuracy and precision, 97.04% recall, 97.03% F-measure, 99.43% AUC, and 0.138 validation loss with twofold cross-validation. When the authors applied fivefold cross-validation, they achieved 98.78% accuracy and 99.88% AUC with a 0.042 validation loss. The results for tenfold cross-validation show that the proposed model achieved 99.22% accuracy and 0.034 loss. The proposed model performs well even with Kfolds. Training and validation accuracy curves using twofold, fivefold and tenfold validation are shown in Fig. 8A and training and validation loss curves using twofold, fivefold and tenfold cross validation are shown in Fig. 8B.

Table 9 K fold cross validation results of the proposed method using twofold, fivefold and tenfold.

Folds	Accuracy	Precision	Recall	F-measure	AUC	Loss	
2	97.04	97.09	97.04	97.03	99.43	0.138	
5	98.78	98.78	98.78	98.79	99.88	0.042	
10	99.22	99.22	99.22	99.25	99.87	0.034	

Figure 8 Results of twofold, fivefold and tenfold cross validation for the proposed model (A) training and validation accuracy curves using twofold, fivefold and tenfold cross validation. (B) Training and validation Loss curves using twofold, fivefold and tenfold cross validation.

Table 10 shows the epoch-wise accuracy of the proposed model. The first ten epochs accuracy and the last ten epochs accuracy, as well as their standard deviation (STD), are calculated. The first ten epochs show that epoch 1 has the lowest accuracy and then increases dramatically at each epoch. The highest accuracy is at epoch 7, which is 99.39. The average accuracy of the first ten epochs is between 0.96 and 0.025 STD. The average accuracy of the last ten epochs is between 99.10 and 0.001 STD.

Comparative analysis of proposed model using Kvasir and WCE datasets with the previous studies

Analysis of the proposed model using the Kvasir and WCE datasets in comparison to previous research is shown in Table 11. When employing an 8-class dataset, the CNN was able to obtain an accuracy of 94.83% (Wang et al., 2022). The DenseNet model achieved accuracy values of 95.63%, precision scores of 89.5%, recall scores of 83.5%, and F-measure scores of 82.5% respectively. In order to classify the diseases, they used images obtained via wireless capsule endoscopy (Hosain et al., 2022) . The authors employed 5-class dataset for the detection of gastointesrinal disorders, and they attained an accuracy rate of 96% (Noor et al., 2023). The study Owais et al. (2019) demonstrated a deep learning-based classification-driven retrieval technique. This approach might identify gastrointestinal diseases. A deep learning classification network and identification technique comprise the structure. The classification network analyzes the current medical status to make an informed prediction about the disease, and the retrieval component displays relevant instances from the database. Khan et al. (2022) employed Image-Net weights to fine-tune the MobileNetV2 model for gastrointestinal diseases. Grad CAM and ResNet152 were also employed by the authors to create a comprehensible deep learning system for endoscopic image classification. They used the free Kvasir database’s 8,000 wireless capsule images.

Table 10 Epoch-wise accuracy of the proposed method.

First 10 Epochs	Last 10 Epochs	
Epochs	Accuracy	Epochs	Accuracy	
1	0.8933	1	0.9922	
2	0.9383	2	0.9922	
3	0.9567	3	0.9922	
4	0.9711	4	0.9894	
5	0.9611	5	0.9906	
6	0.9728	6	0.9906	
7	0.9783	7	0.9939	
8	0.9739	8	0.9922	
9	0.9806	9	0.9872	
10	0.9811	10	0.9894	
Average	0.9607	Average	0.9910	
STD	0.0257	STD	0.0018	

Table 11 Analysis of the proposed method in comparison to other approaches.

Methods	Diseases	Performance metrics	
		Accuracy	Precision	Recall	F-measure	AUC	Loss	
CNN (Wang et al., 2022)	Eight diseases	94.83%	–	–	–	–	–	
DenseNet201 (Hosain et al., 2022)	Four diseases	95.63%	89.5%	83.5	82.56	–	–	
MobileNetV2 (Noor et al., 2023)	Five diseases	96.40	93.02	97.57	95.24	–	–	
CNN+LSTM (Owais et al., 2019)	37 diseases	92.57%	93.41	–	–	97.05	–	
Densenet, LSTM and KNN (Fati, Senan & Azar, 2022)	37 diseases	96.19	98.18	95.86	96.99	–	–	
CNN (Iftikhar et al., 2017)	Five diseases	96.65	95.41	96.06	95.74	–	–	
SSD (Xiao et al., 2023)	Four diseases	97.25	–	–	–	–	–	
CNN (Mukhtorov et al., 2023)	Eight diseases	93.46	–	–	–	–	0.11	
VGG16 (Dheir & Abu-Naser, 2022)	Eight diseases	98.2	–	–	–	–	0.23	
Proposed method	Eight diseases	99.22	99.22	99.22	99.25	99.87	0.03	
	Four diseases	96.63	98	97	97	99.58	0.11	

An effective augmentation technique was employed to classify medical images using the heat map of classification results, which had an accuracy of 98.2% during training and 93.46% during validation (Mukhtorov et al., 2023). The previous results on Gastrointestinal tracts demonstrate that the proposed model is outclassed in terms of all performance metrics; it achieved 99.22% accuracy on dataset 1 (eight classes) and 96.63% on dataset 2 (four classes). Also, the proposed model achieved 99.87% and 99.58% AUC, respectively.

Deep learning and optimized EfficientNet models with CNN layers are used in the experimental design to make the distribution of dataset classes more fair. Data augmentation is a practice that artificially increases the size of the training set by creating updated copies of the dataset using existing data. The optimized model, which incorporates the augmentation approach, has proven to be more beneficial in improving model performance. The optimized model achieved the highest level of accuracy, with a quicker processing time of 44 s per epoch compared to the other model. The model’s accuracy ranged from 0.96 to 0.025 standard deviations over the first ten epochs, with the highest accuracy at 99.39 at the seventh epoch. Validation of the model was carried out using the twofold, fivefold, and tenfold methods. The authors achieved 97.04% accuracy and precision with twofolds, 98.78% accuracy with fivefolds, and 99.22% accuracy with tenfold. Despite the presence of Kfolds, the suggested model functions admirably, demonstrating that augmentation strategies are more beneficial for increasing model performance.

RQ1: The choice of techniques for data enhancement significantly improves the performance of an optimized EfficientNet model in the classification of gastrointestinal diseases. Effective use of augmentation techniques like as flipping, rotating, and zooming can increase the model’s ability to generalize and reduce overfitting. As a result, the network can better extract distinctive features from the enhanced data. Therefore, optimizing the augmentation pipeline is critical to finding a balance that boosts the model’s generality while limiting detrimental implications on its capacity to correctly recognize gastrointestinal tracts.

RQ2: The proposed model attained maximum accuracy while requiring few computational resources. The proposed model has a much faster processing time of 44 s per epoch, in contrast to the Xception model, which takes 72 s per epoch. The model’s underlying parameterization enables easy scalability, making it an excellent choice for scenarios with limited resources. The integration of efficiency, cost-effectiveness, and parameter optimization makes it a robust and adaptable solution for classifying gastrointestinal tracts. It meets the demands of real-world applications while maintaining high levels of accuracy and reliability.

RQ3: We validate the efficacy of our proposed model by applying it to a separate dataset for the purpose of classifying gastrointestinal diseases. The results obtained were outstanding, further demonstrating the usefulness of our model. The testing demonstrated its reliability as a versatile method for efficiently and correctly detecting diseases in diverse healthcare environments.

Conclusion and Future work

This study proposes an enhanced EfficientNet framework for detecting Gastrointestinal disorders using Transfer Learning and Wireless Capsule Endoscopy Images. Deep learning algorithms take time and computational resources to train huge datasets, according to previous studies. We solve these limitations by optimizing the EfficientNet classification model in our study. The model uses transfer learning and fine-tuning, and is tested on a separate endoscopic dataset to ensure its lifespan and dependability, overcoming limitations in training large datasets. The research was conducted with two publicly available datasets. The method identifies the best features to increase classification accuracy while reducing computational cost. Experiments showed that the pretarined models such as VGG16 achieved 97.61% accuracy, CNN achieved 95.22% accuracy, ResNet50 achieved 96.61% accuracy, Xception achieved 77.94% accuracy, and DenseNet121 achieved 88.11% accuracy using the Kvasir dataset. The proposed model achieved 99.22% accuracy and 99.87% area under the curve-score on dataset 1 and 99.58% AUC-score on dataset 2. The proposed model achieved the best validation loss of 0.034, which is lower than other models. We also compare the computational time of the proposed model with other models. The proposed model takes only 44 s per epoch during the training process, is more efficient and reliable, and produces high performance in early diagnosis. The proposed model provides assistance with an artificial intelligence system to endoscopic professionals.

In the future, intensive testing will be carried out to improve the precision of deep feature maps derived from deep learning models. The method of feature removal and the PCA algorithm will be employed to reduce the dimensionality of these maps. The dataset will be expanded to incorporate more than eight classifications, and clinical trials will be carried out to assess the efficacy of the model in detecting gastrointestinal diseases.

Supplemental Information

Supplemental Information 1 Experimental code

Abbreviations

WHO World Health Organization

IB intestinal biopsy

DCNN Deep Convolutional Neural Networks

TL Transfer learning

VT Vision transformer

LSTM Long short term memory

VGG Visual Graphics Group

AUC Area under the curve

Mo model

CNN Convolutional Neural Networks

TP True Positive

DL Deep Learning

Conv2D Convolutional 2 dimensional

ReLU Rectified Linear Unit

Additional Information and Declarations

Competing Interests

Author Contributions

Data Availability

The authors declare there are no competing interests.

Shaha Al-Otaibi performed the experiments, performed the computation work, authored or reviewed drafts of the article, and approved the final draft.

Amjad Rehman conceived and designed the experiments, analyzed the data, prepared figures and/or tables, and approved the final draft.

Muhammad Mujahid conceived and designed the experiments, performed the experiments, analyzed the data, performed the computation work, prepared figures and/or tables, authored or reviewed drafts of the article, and approved the final draft.

Sarah Alotaibi performed the computation work, prepared figures and/or tables, and approved the final draft.

Tanzila Saba analyzed the data, performed the computation work, prepared figures and/or tables, and approved the final draft.

The following information was supplied regarding data availability:

Dataset 1: Kvasir: A multi-class image dataset for computer aided gastrointestinal disease detection. https://datasets.simula.no/kvasir/

Dataset 2: WCE Curated Disease. https://www.kaggle.com/datasets/francismon/curated-colon-dataset-for-deep-learning

The code is available in the Supplemental File.

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
