# Peer review of "Efficient-gastro: optimized EfficientNet model for the detection of gastrointestinal disorders using transfer learning and wireless capsule endoscopy images"

_PeerJ Computer Science, doi:10.7717/peerj-cs.1902_

## Round 0.1 · original submission · Major Revisions

With reviewers comments provide more information about datasets and results

**Language Note:** The review process has identified that the English language must be improved. PeerJ can provide language editing services - please contact us at copyediting@peerj.com for pricing (be sure to provide your manuscript number and title). Alternatively, you should make your own arrangements to improve the language quality and provide details in your response letter. – PeerJ Staff

Reviewer 1 ·

Basic reporting

English and Writing Quality

The paper is well-written overall with clear, professional English. Just some minor grammatical issues to address.
Be sure to carefully proofread to fix any lingering typos, grammar issues, or awkward phrasings.
Define all acronyms on first use (e.g. CNN, AUC) and reduce acronym usage for readability.
Figures use inconsistent styles and could be higher quality. Ensure all visuals meet publication standards.
Literature References and Background

The introduction and background provide good context, but could go into more detail on specific knowledge gaps in prior art that this work aims to address.
Further emphasize what new contributions this work makes by comparing to limitations of existing methods in the literature review section.
Article Structure and Organization

The paper's structure generally follows conventions, but some rearrangement of sections may improve narrative flow.
Figure quality and table formatting could be refined to more clearly present key results.
Data and Results

Provide more details on the datasets used - their size, annotation process, class balance, etc. This context aids reproducibility.
The results analysis is appropriate and addresses the initial research questions and hypotheses.
Sharing code and data allows readers to further validate the fidelity of findings. Ensure public access.
Formalization of Methods and Results

More implementation details are needed around model parameters, training configurations, data splits etc. to enable reproducibility.
Provide clear definitions of quantities analyzed in the results like accuracy, AUC etc.

Experimental design

Research Question and Contributions

The authors clearly articulate the challenges and knowledge gaps in automated gastrointestinal disease detection, situating the motivation for the work.
However, the specific research questions and tangible contributions of this work could be more explicitly stated in the introduction.
Highlight how this work builds on and advances prior state-of-the-art in more detail. What limitations does it address?
Investigation Rigor

Appropriate datasets are utilized and the data collection methodology is sound, relying on medical experts for annotations. This lends credibility.
Preprocessing, augmentation, modeling and evaluation steps represent technically solid investigative practices that follow community norms.
Some discussion of ethical considerations around patient data use would further strengthen rigor.
Methodological Detail

While major methods are well-summarized, more granular details around implementations would allow direct reproducibility.
Provide specific model architecture details, parameter values, training configurations, computational requirements, and data splits and pre-processing code.
Analysis relies on standard metrics but include mathematical definitions of accuracy, AUC etc. for completeness.

Validity of the findings

Impact and Novelty

The paper could better highlight the novelty of the proposed approach and its impact for gastrointestinal disease detection.
How specifically does it push boundaries beyond prior state-of-the-art? Comparisons on limitations addressed would strengthen perceived novelty and importance.
Replication is meaningful because of the technical value and potential healthcare impacts detailed.
Data Robustness

The use of expert-annotated public benchmark datasets suggests the underlying data is reliable and representative.
Additionally sharing code and raw results allows independent validation and controls for quality.
More details on dataset balance across classes could further validate statistical soundness.
Conclusions

The conclusions generally summarize experimental results, link back to original goals & questions.
But discussions of limitations and open challenges are brief. Further qualify conclusions - what still remains to be addressed?
Take care not to overstate findings beyond what the restricted experimental scope fully supports. Bound claims to evidence.

Additional comments

Minor Comments:

Reduce acronym usage for readability, or include a table of acronyms defined.
Carefully proofread the paper - there are several grammatical issues.
Figure quality could be improved - higher resolution, more informative captions, consistent style.
References are comprehensive but the citation style is inconsistent.
Section organization could be refined - some rearrangement may improve narrative flow.

Reviewer 2 ·

Basic reporting

1. The abstract is too wordy. It would be better to squeeze it for more meaningful and efficient communication.
2. The motivation behind the proposed scheme should be included in the introduction section, prior to the contributions.
3. There is no paper organization mentioned. I recommend adding a section on paper organization at the end of the introduction.
4. The related work section uses present tense (e.g., Escobar et al. (2021) published, Hosain et al. (2022) employed). This should be revised to past tense. Additionally, some sentences are informal (HyperKvasir data, according to the authors; The paper Hosain et al. (2022); Authors in study Nouman Noor et al. (2023); The study Saba et al. (2018) demonstrated; The findings of the developed model was fully analyzed; Khan et al. (2022) in another publication employed various preprocessing techniques). These should be formalized.
5. The paper should provide detailed information on the specific model architecture, parameter values, and training configurations.
6. A preliminaries section is needed to clearly define the quantities analyzed in the results.
7. More comprehensive details on the datasets used are required, as the current information is insufficient. Also, the performance analysis section should be squeezed.
8. The limitations of the proposed scheme should be clearly stated. It would be better to include future work.

Experimental design

All the comments are added in section 1. Basic reporting

Validity of the findings

All the comments are added in section 1. Basic reporting

Additional comments

All the comments are added in section 1. Basic reporting

---

## Round 0.2 · accepted · Accept

The paper is now acceptable based on the reviewers' recommendations.

Reviewer 1 ·

Basic reporting

Yes, the authors have significantly improved the paper.

Experimental design

Yes, the authors have significantly improved the experimental design and elaborated every comment in detail.

Validity of the findings

Validity of the findings is good and all the comments are addressed.